# From the Physicochemical Characteristic of Novel Hesperetin Hydrazone to Its In Vitro Antimicrobial Aspects

**DOI:** 10.3390/molecules27030845

**Published:** 2022-01-27

**Authors:** Anna Sykuła, Elżbieta Łodyga-Chruścińska, Eugenio Garribba, Dorota Kręgiel, Aliaksandr Dzeikala, Elżbieta Klewicka, Lidia Piekarska-Radzik

**Affiliations:** 1Institute of Natural Products and Cosmetics, Faculty of Biotechnology and Food Sciences, Lodz University of Technology, 90-537 Łódź, Poland; elzbieta.lodyga-chruscinska@p.lodz.pl (E.Ł.-C.); aliaksandr.dzeikala@gmail.com (A.D.); 2Department of Medical, Surgical and Experimental Sciences, University of Sassari, Viale San Pietro, I-07100 Sassari, Italy; garribba@uniss.it; 3Department of Environmental Biotechnology, Faculty of Biotechnology and Food Sciences, Lodz University of Technology, 90-530 Łódź, Poland; dorota.kregiel@p.lodz.pl; 4Institute of Fermentation Technology and Microbiology, Faculty of Biotechnology and Food Sciences, Lodz University of Technology, 90-530 Łódź, Poland; elzbieta.klewicka@p.lodz.pl (E.K.); lidia.piekarska-radzik@edu.p.lodz.pl (L.P.-R.)

**Keywords:** antibiofilm molecules, Schiff bases, copper complexes, antibacterial activities, antiadhesive action

## Abstract

Microorganisms are able to give rise to biofilm formation on food matrixes and along food industry infrastructures or medical equipment. This growth may be reduced by the application of molecules preventing bacterial adhesion on these surfaces. A new Schiff base ligand, derivative of hesperetin, HABH (2-amino-N′-(2,3-dihydro-5,7-dihydroxy-2-(3-hydroxy-4-methoxyphenyl)chromen-4-ylidene)benzohydrazide), and its copper complex, CuHABH [CuLH_2_(OAc)], were designed, synthesized and analyzed in terms of their structure and physicochemical properties, and tested as antibacterial agents. Their structures both in a solid state and in solution were established using several methods: FT-IR, ^1^H NMR, ^13^C NMR, UV-Vis, FAB MS, EPR, ESI-MS and potentiometry. Coordination binding of the copper(II) complex dominating at the physiological pH region in the solution was found to be the same as that detected in the solid state. Furthermore, the interaction between the HABH and CuHABH with calf-thymus DNA (CT-DNA) were investigated. These interactions were tracked by UV-Vis, CD (circular dichroism) and spectrofluorimetry. The results indicate a stronger interaction of the CuHABH with the CT-DNA than the HABH. It can be assumed that the nature of the interactions is of the intercalating type, but in the high concentration range, the complex can bind to the DNA externally to phosphate residues or to a minor/major groove. The prepared compounds possess antibacterial and antibiofilm activities against Gram-positive and Gram-negative bacteria. Their antagonistic activity depends on the factor-strain test system. The glass was selected as a model surface for the experiments on antibiofilm activity. The adhesion of bacterial cells to the glass surface in the presence of the compounds was traced by luminometry and the best antiadhesive action against both bacterial strains was detected for the CuHABH complex. This molecule may play a crucial role in disrupting exopolymers (DNA/proteins) in biofilm formation and can be used to prevent bacterial adhesion especially on glass equipment.

## 1. Introduction

Diverse microorganisms are able to grow on food matrixes and along food industry infrastructures and this growth may give rise to biofilms [1]. Knowledge of the bacterial species responsible for the pre-colonization, maturation and dispersal of biofilms in the food industry is very important due to the health problems associated with, for example, dairy products, ready-to-eat foods or alimentary goods [1]. These human pathogens include, but are not limited to, *Escherichia coli* (which may include enterotoxic and even hemorrhagic strains), *Listeria monocytogenes* (a species ubiquitous in soil and water that can lead to abortion in pregnant women and other serious complications in children and the elderly), and *Staphylococcus aureus* (known to have numerous intestinal toxins) [1,2,3].

The pathogens are able to develop biofilm structures on different artificial substrates common in the food industry, such as glass, stainless steel, polyethylene, wood, polypropylene, rubber, etc. Biofilm formation represents a special mode of growth that renders microbial cells more resistant to antimicrobials enabling the pathogens to survive in hostile environments and also to disperse and colonize new niches. Numerous studies have analyzed the possible relationship between biofilm formation and antimicrobial resistance [4,5,6].

Microbial resistance is multifactorial; there are multiple mechanisms of resistance that act together in order to provide an increased overall level of resistance to the biofilm. These mechanisms are based on the function of wild-type genes and are not the result of mutations [7].

Bacterial species that make up the biofilms show genomic differences with respect to key genes involved in biofilm characterization, leading to the formation of completely different biofilms under different conditions [8]. This complexity, together with the great variety of environments affected and the diversity of bacterial colonizing species, complicates biofilm removal in the food industry.

The global spread of antibiotic resistance among important human pathogens emphasizes the need to find new antibacterial drugs with a novel mode of action. These new antibiofilm agents, which contain moieties such as phenols, imidazole, sulfide, furanone, etc., have the potential to disperse bacterial biofilms in vivo and could positively impact human medicine in the future. Nature continues to inspire the discovery of new compounds with interesting structures and biological activities and these naturally-derived compounds have served as scaffolds for the development of many synthetic therapeutic agents. This has prompted us to investigate derivatives of natural flavonoids.

Flavonoids are considered as natural therapeutics playing major roles in successful medical treatments in ancient as well as modern times [9]. They belong to the group of compounds inhibiting biofilm formation [10]. We have undertaken research on hesperetin, one of the representatives of flavonoids belonging to the flavanone subgroup. On the one hand, Schiff’s bases are well known as active compounds and exhibit a range of activities, including antimicrobial activity [11,12,13]. Moreover, the Schiff bases’ metal complexes have been of major interest for a long time because of their more significant antimicrobial potency [14,15,16]. This inspired us to take up the challenge of synthesizing the Schiff bases of flavanone and we have documented that hesperetin or naringenin bonded to the azomethine group associated with active benzohydrazide or thiosemicarbazide moieties that showed a DNA damaging potency and antibacterial activity [17,18].

In the present work we have focused on the research of a novel compound (HABH) synthesized by the condensation of 2-aminobenzhydrazide and hesperetin (Figure 1) and its copper(II) complex (CuHABH). Their structure and physicochemical properties were characterized by spectroscopic (FT-IR, ^1^H NMR, ^13^C NMR, UV-Vis, FAB MS, EPR, ESI-MS) and electrochemical techniques. These compounds were screened for *in vitro* antibacterial activity against seven different bacterial strains and for antibiofilm activities against Gram-positive *Staphylococcus aureus* and Gram-negative *Escherichia coli* strains.

Adhesion to the surfaces of inert materials, such as glass or plastics, has been recognized as resulting from physicochemical interactions between different components (microorganism, substrate and suspending medium). Under the biofilm state, bacteria produce the extracellular polymeric substances in order to build up the matrix that holds the sessile cells together. Recently, extracellular exopolysaccharides, extracellular DNA and proteins [19,20,21] have been found as the main components of the biofilm matrix. With this in mind, we have made an attempt to establish the interactions of HABH and CuHABH with CT DNA, the molecule used in model studies of the interaction of compounds with potential biological activity. These interactions have been tracked by spectrofluorimetry. Adhesion of the bacterial cells to the glass surface in the presence of the compounds were traced by luminometry.

## 2. Materials and Methods

### 2.1. Materials in the Synthesis of Novel Hesperetin Derivatives

The racemic hesperetin, 2-aminobezhydrazide, and all other compounds were purchased from Sigma-Aldrich Co. (Poznań, Poland). All reagents were of an analytical quality and were used without further purification.

### 2.2. Apparatus

The structure of the compounds obtained was determined by: elemental analysis (C, H, N) on the EuroVector 3018 analyzer (EuroVector, Milan, Italy); analysis of the FAB-MS mass spectra made on a Finnigan MAT 95 mass spectrometer (Finnigan MAT, San Jose, CA, USA); and analysis of the IR spectra using an FT-IR spectrometer Nicolet 6700 (Thermo Scientific, Waltham, MA, USA) in the range of 4000–350 cm^−1^. The melting point of the ligands was determined on an Electrothermal 9200 microscopic melting point apparatus (Cole-Parmer Ltd., Stone, Staffordshire, UK). ^1^H NMR spectra analyses were also performed on a Bruker Avance III WB 400-MHz spectrometer (Bruker, Ettlingen, Germany) in DMSO-d_6_ with tetramethylsilane (TMS) as an internal standard. Chemical shifts were expressed in parts per million (ppm, δ) and were referenced to DMSO-d_6_ (δ = 2.50 ppm) as an internal standard. The ^13^C NMR spectra were also obtained using a Bruker Avance III 400 spectrometer (Bruker, Ettlingen, Germany) that operated at 100 MHz. Chemical shifts were expressed in parts per million (ppm, δ) and were referenced to DMSO-d_6_ (δ = 39.52 ppm) as an internal standard. All NMR spectra were taken in DMSO-d_6_ (Deutero Gmbh, Kastellaun, Denmark). UV-Vis spectra of the compounds in the DMSO and in the mixture of 70% HCl(NaCl)/30% were recorded in the λ interval 200–900 nm using a Hewlett-Packard 8453 spectrophotometer running 845x UV-Visible ChemStation Software (Agilent, Mulgrave, Victoria, Australia). Solutions were inserted in a quartz cell with a path length of 1 cm. The content of copper in the complex was determined by a 932 AA spectrophotometer from GBC (Dandenong, VIC, Australia) with a deuterium background correction used. The system was controlled by a data station running the GBC Avanta Version 1.33 (GBC, 932 AA spectrophotometer, Dandenong, VIC, Australia).

The formation of the new compounds was monitored by electrospray ionization mass spectrometry (ESI-MS, Bruker Daltonics GmbH, Bremen, Germany) analysis in the negative ion mode. Before the analysis, the samples were dissolved in acetonitrile (250 µg/mL) and injected directly into a Q-Exactive Orbitrap™ (Thermo Scientific, Bellefonte, PA, USA) tandem mass spectrometer equipped with a heated electrospray ionization (ESI) interface (HESI–II), using an injection pump and a 500 µL syringe (Thermo Scientific, Bellefonte, PA, USA). The injection speed was 10 µL/min. The new compounds were analyzed using full-scan MS and a subsequent parallel reaction monitoring (PRM, Thermo Scientific, Hudson, NH, USA) mode with a scan range from 50 to 1000 *m/z*. The capillary temperature was adjusted to 320 °C. The electrospray capillary voltage and S-lens radio frequency (RF) level were set at 4.5 kV and 50 V, respectively. Nitrogen was used as both a sheath gas and auxiliary gas at a flow of 10 and 8 (arbitrary units), respectively. Ions that were selected by the quadrupole entered the higher energy collision dissociation (HCD) cell. An isolation window of 2 amu was used and the precursors were fragmented by a collision-induced dissociation C-trap (CID) with a normalized collision energy (NCE) of 25 V. The ESI-MS/MS scan spectra (Bruker Daltonics GmbH, Bremen, Germany) were acquired with the mass resolution of 35,000 full-width at half-maximum (FWHM) at *m/z* = 100. The automatic gain control (AGC) target (the number of ions to fill C-Trap) was set at 2.0 × 10^5^ with a maximum injection time (IT) of 100 ms. The instrument control, data acquisition, and evaluation were completed with the Q Exactive Tune 2.1 and Thermo Xcalibur 2.2 software (Thermo Fisher Scientific, Bremen, Germany).

#### 2.2.1. Characterization of HABH

HABH synthesis is presented in Figure 1.

The ligand 2-amino-benzoic acid [5,7-dihydroxy-2-(3-hydroxy-4-methoxy-phenyl)-chroman-4-ylidene]-hydrazide (HABH) was prepared according to the literature procedure [22,23] with some modifications. The synthesis product was filtered off, washed with ethanol and methanol and then dried to give a solid ligand. The synthesis yield was 67%.

The 2-amino-benzoic acid [5,7-dihydroxy-2-(3-hydroxy-4-methoxy-phenyl)-chroman-4-ylidene]-hydrazide (HABH) was characterized by using several spectroscopic methods. Yield: 891 mg, 67%; m.p. 264–266.5 ^0^C; ^1^H NMR (DMSO-d_6_, 400 MHz) δ: 2.92 (dd, *J*_1_ = 12.1 Hz, *J*_2_ = 17.1 Hz, 1H), 3.39 (dd, *J*_1_ = 3.0 Hz, *J*_2_ = 17.2 Hz, 1H), 3.35 (s, 3H), 5.09 (dd, *J*_1_ = 2.9 Hz, *J*_2_ = 12.3 Hz, 1H), 5.87 (d, *J* = 2.2 Hz, 1H), 5.93 (d, *J* = 2.3 Hz, 1H), 6.33 (bs, 1H), 6.53 (td, *J*_1_ = 0.9 Hz, *J*_2_ = 7.8 Hz, 1H), 6.74 (d, *J* = 7.8 Hz, 1H), 6.87–6.96 (m, 3H), 7.10 (td, *J*_1_ = 1.4 Hz, *J*_2_ = 8.1 Hz, 1H), 7.53 (d, *J* = 6.5 Hz, 1H), 9.11 (s, 1H), 9.97 (s, 1H), 10.80 (bs, 1H), and 13.16 (s, 1H); ^13^C NMR (DMSO-d_6_, 100 MHz) δ: 32.1, 55.7, 75.9, 94.9, 96.7, 98.4, 112.0, 113.4, 113.9, 114.6, 116.2, 117.5, 129.1, 132.1, 132.4, 146.5, 147.7, 149.8, 153.5, 158.9, 160.8, and 161.3 (^1^H NMR and ^13^C spectra of HABH are available as Appendix A); IR *ν*_max_(cm^−1^): *ν*(O-H): 3421, *ν*(N-H): 3333, *ν*(C=O): 1630, *ν*(C=N): 1609, *ν*(C=C): 1530, *ν*(C–O): 1253, *ν*(C–O–C): 1150, *ν*(C–N): 1017, and *ν*(N–N): 1020; UV–Vis λ_max_(nm): 323, 349 nm; FAB-MS: *m/z* = 436 [M + H]^+^; Anal. Calc. for C_23_H_21_N_3_O_6_: C, 63.44; H, 4.86; N, 9.65. Found: C, 63.39; H, 4.89; and N, 9.59.

#### 2.2.2. Characterization of Copper(II) Complex with HABH in Solid State

An amount of 72.58 mg (0.1169 mmol) of hesperetin 2-aminobenzoylhydrazone (HABH) was quantitatively transferred to a round bottom flask and dissolved in 15 mL of acetone at 60 °C. The contents of the flask were stirred for 10 min until the HABH was completely dissolved, then 35.60 mg (0.1783 mmol) of Cu(OAc)_2_·xH_2_O was added to the obtained solution, with one drop of triethylamine added as a catalyst and the reaction was carried out at 60 °C within 5 h. After this time, a dark green copper complex precipitated, which was suction filtered, washed with acetone and methanol and left to dry in the desiccator. The synthesis yield was 67%.

For the [CuLH_2_(OAc)] (CuHABH as abbreviation); Yield: 58.57 mg, 63%; Anal. Calc. C_25_H_23_CuN_3_O_8_: C, 53.91; H, 4.16; N, 7.54; Cu, 11.41. Found: C, 53.63; H, 4.09; N, 7.48; Cu, 11.04%. IR ν_max_ (cm^−1^): *ν*(N–H): 3395, *ν*(C=O): 1613, *ν*(C=N): 1593, *ν_as_*(COO^-^): 1563, *ν*(C=C): 1436, *ν_s_*(COO^-^): 1363, *ν*(C–O): 1278 *ν*(C–O–C): 1159, *ν*(C–N): 1068, *ν*(N–N): 1031, *ν*(M–O): 445, *ν*(M–N): 410. UV–Vis λ_max_ (nm): 284, 338 for 70% HCl(NaCl)/30% DMSO and 335; 289; 406 for 100% DMSO. ESI–MS: *m*/*z* = 555.38 [CuLH_2_(OAc)][M-H]^−^ (Mass spectrum is available as Appendix A).

#### 2.2.3. Stoichiometry Determination of the Copper(II) Complex with HABH in Solution

##### Potentiometry

Complex stoichiometry was determined by pH-potentiometric titrations of 2.0 mL samples in DMSO/water mixtures (30%:70% *v*/*v*) due to the slight solubility of the CuHABH in pure water (Appendix A) and the UV-Vis (Appendix A) and EPR studies. The ligand:metal molar ratio was 1:1, the concentrations of Cu(II) and the ligand were 1 × 10^−3^ M. The measurements were carried out at 298 K and at a constant ionic strength of 0.1 M KCl with a MOLSPIN pH meter (Molspin Ltd., Newcastle-upon-Tyne, UK), equipped with a digitally operated syringe (the Molspin DSI 0.250 mL) controlled by computer. The titrations were performed with a carbonate-free NaOH solution of a known concentration (ca. 0.1 M) using a Russel CMAWL/S7 semi-micro combined electrode. The pH measuring circuit was calibrated with potassium hydrogen phthalate and phosphate buffers. The number of experimental points was 100–150 for each titration curve. The reproducibility of the titration points included in the evaluation was within 0.005 pH units in the whole pH range examined (2.0–11.5). Complex stoichiometry and diagram distributions of the protonated/deprotonated species were revealed (Appendix A) using the computer program SUPERQUAD [24]. The procedure of the calculations was adopted from our previous works [17,18].

##### Spectroscopy and Computational Details

The presence of protonated/deprotonated species was supported by electronic absorption (UV–Vis) spectra recorded in the DMSO/water mixture (30%/70% v/v fraction) at a different pH corresponding to the dominant species of the HABH or CuHABH. C_HABH_ = C_CuHABH_ = 1 × 10^−5^ M. Spectra were recorded with a Perkin-Elmer Lambda 11 spectrophotometer (Perkin Elmer, Ueberlingen, Germany) using a quartz cell with a path length of 1 cm (Appendix A). Coordination modes of the copper complexes with the HABH were studied by electron paramagnetic resonance (EPR) spectroscopy. The spectra were recorded from 0 to 8000 Gauss at room temperature (298 K) or liquid nitrogen temperature (77 K) with an X-band Bruker EMX spectrometer (Bruker Corporation, Billerica, MA, USA) equipped with a HP 53150A microwave frequency counter. The spectra in an aqueous solution as a function of the pH were recorded on the system ^63^CuSO_4_·5H_2_O/H_4_ABH 1:1 with a Cu^II^ concentration of 1.0 × 10^–3^ M. The ^63^CuSO_4_·5H_2_O, used for a better resolution of the EPR spectra, was prepared from metallic copper (99.3% ^63^Cu and 0.7% ^65^Cu) purchased from JV Isoflex (Isoflex, Moscow, Russian Federation). The microwave frequency used to record the spectra was in the range 9.40–9.42 GHz, microwave power was 20 mW, the time constant was 163.8 ms, modulation frequency 100 kHz, modulation amplitude 4 Gauss, sweep time 335.5 s, and the resolution 2048 (range 2500–3500 Gauss) or 8192 (range 0–8000 Gauss) points. To extract the experimental spin Hamiltonian parameters, WinEPR SimFonia (Bruker Analytische Messtechnik GmbH, Karlshrue, Germany) software was used [25]. The geometry was optimized, and harmonic frequencies were computed with Gaussian 09 software [26] at the DFT theory level. The hybrid Becke three-parameter B3LYP functional [27,28], coupled with the Grimme’s D3 dispersion was used combined with the split-valence Pople basis set 6-311g(d,p) for the main group elements, while the Stuttgard-Dresden (SDD) implemented with *f*-functions and pseudo-potential was applied for the Cu. These conditions have been successfully applied and discussed in the literature for the geometry prediction of first-row transition metal complexes [29,30,31]. The ***g*** and ***A*** tensors of the ^63^Cu center were calculated with the ORCA program [32,33,34], using the functionals PBE0 [35,36] and B3LYP [27,28], respectively, as recently established in the literature [37].

### 2.3. DNA Studies

#### 2.3.1. Electronic Absorption Titration

Electronic absorption spectra of the compounds HABH, CuHABH and HESP (25 μM) were recorded with a gradual increasing concentration of CT-DNA (0–63 μM). The intrinsic binding constant (*K_b_*) of the complex with the CT-DNA was determined using the following equation [38] (Equation (1)): (1)$$\left[DNA\right]/\left(\varepsilon_{a}-\varepsilon_{f}\right)=\left[DNA\right]/\left(\varepsilon_{b}-\varepsilon_{f}\right)+1/[K_{b}\left(\varepsilon_{b}-\varepsilon_{f}\right)]\;$$
where [*DNA*] is the concentration of CT-DNA, *ε_a_* is the molar extinction coefficient of the compound at a given CT-DNA concentration, and *ε_f_* and *ε_b_* are the extinction coefficients of the compound in free solution and when fully bound to the CT-DNA, respectively. The plot of [*DNA*]/(*ε_a_* − *ε_f_*) vs. [*DNA*] gives a straight line with 1/(*ε_b_* − *ε_f_*) and 1/[*K_b_*(*ε_b_* − *ε_f_*)] as the slope and intercept, respectively. From the ratio of the slope to the intercept, the value of *K_b_* was calculated.

#### 2.3.2. Competitive Binding Fluorescence Measurement

The thiazole orange (TO)-competitive studies of each compound were carried out with fluorescence spectroscopy to examine whether the compounds tested can replace TO in the TO bounded CT-DNA system. An aqueous solution of TO (25 μM) bounded to the CT-DNA (25 μM) was prepared in a Tris-HCl(NaCl) buffer (pH 7.2) and this was titrated with solutions of the tested compounds (concentrations in the range of 10–300 µM). In the presence of CT-DNA, thiazole orange (TO) exhibits a fluorescence (λ_em_ = 527 nm, λ_ex_ = 509 nm) [39] enhancement due to its intercalative binding to DNA. Competitive binding of compounds with the CT-DNA results in fluorescence quenching due to the displacement of TO from CT-DNA. The Stern–Volmer constant (*K_SV_*) was calculated using the Stern–Volmer equation [40,41] (Equation (2)):(2)$$I_{0}/I=1+K_{SV}\left[Q\right]$$
where *I*_0_/*I* is the fluorescence quenching index, *I*_0_ and *I* are the fluorescence emission intensities in the absence and presence of the quencher (tested compound), respectively, *K_SV_* is the Stern–Volmer constant and [*Q*] is the concentration of the quencher. Taking *τ_o_* = 2.6 ns as the fluorescence lifetime of the TO-DNA adduct [42], the quenching constants (*k_q_*, in M^−1^s^−1^) of the compounds were calculated according to Equation (3) [40]:(3)$$K_{SV}=k_{q}\tau_{o}$$

#### 2.3.3. Circular Dichroism Spectroscopy Analysis

Circular dichroism (CD) spectra of the CT DNA with the HABH and CuHABH were recorded using the Jasco J815 CD spectropolarimeter (Jasco, Easton, MD, USA) in the wavelength range (λ) 200–460 nm at room temperature (298 K) using quartz cuvettes with a 0.5 cm optical path, applied bandwidth of 2 nm and integration time of 1–2 s. Compounds in the range of concentration 10.6 µM to 100 µM were dissolved in a DMSO/Tris-HCl solution (5 mM Tris-HCl, 50 mM NaCl, pH 7.2) and added to the CT DNA (100 µM) in a Tris-HCl buffer. The reference sample was a Tris-HCl buffer. The DMSO content in the solutions was <2%.

### 2.4. Antibacterial Activity

The antimicrobial activity of the CuHABH, HABH, and HESP were tested against Gram-positive bacteria: *Staphylococcus aureus* ATCC25923, *Staphylococcus aureus* ATCC27734, *Listeria monocytogenes* ATCC 19111, and *Listeria monocytogenes* ATCC 19115; and Gram-negative bacteria: *Salmonella* Choleraesuis ATCC 7001, *Salmonella* Typhimurium ATCC 14028, and *Escherichia coli* ATCC 10536. The strains were activated from cryobanks by passaging twice onto a nutrient broth (Merck, Darmstadt, Germany). Bacteria *S. aureus* ATCC25927, *S. aureus* ATCC27734, *S*. Choleraesuis ATCC 7001, *S*. Typhimurium ATCC 14028 and *E. coli* ATCC 10536 were grown for 24 h at 37 °C while *L. monocytogenes* ATCC 19111, and *L. monocytogenes* ATCC 19115 were cultured for the same time at 30 °C. Testing of antagonistic activity was performed according to the procedures recommended by the European Committee on Antimicrobial Susceptibility Testing EUCAST [43]. The inoculum suspension was prepared by selecting several colonies from overnight growth on the nutrient agar (Merck, Darmstadt, Germany) medium with a sterile loop and suspending the colonies in a sterile saline (0.9% w/v NaCl in water) to the density of 0.5 McFarland standard (1–2 × 10^8^ CFU/mL). The samples of the test compounds were dissolved in DMSO to obtain a concentration of 5 mg/mL and were sterilized by filtration (filter pore width 0.2 μm; (Sartorius AG, Gottingen, Germany). Paper disks (∅ = 6 mm, Oxoid, (Thermo Fisher Scientific, Waltham, MA, USA)) were impregnated with compound samples, to obtain a concentration of the test compounds of 0.1 μM per disk, and the solvent was allowed to evaporate in the dark at room temperature. A sterile cotton swab was dipped into the inoculum bacterial suspension and the excess fluid removed by turning the swab against the inside of the tube to avoid over-inoculation of the plates. The inoculum was spread evenly over the entire surface of the agar plate by swabbing in three directions. The bacteria inoculum test culture smear was on sterile Mueller–Hinton Agar (Merck, Darmstadt, Germany) plates. The impregnated paper disks were placed on the plates’ surface. DMSO solution was used as a negative control at the concentration of 20 mg/mL (this concentration of DMSO did not inhibit the growth of microorganisms) [44]. The vancomycin (Oxoid Thermo Fisher Scientific, Waltham, MA, USA) was used as the positive control for the Gram-positive bacteria, ampicillin (Oxoid Thermo Fisher Scientific, Waltham, MA, USA) for the Gram-negative bacteria. The plates were incubated at the above-mentioned temperatures for 16–24 h. After the incubation time, the zone inhibition of the bacteria growth was measured. The experiments were performed in two independent replications. The results were expressed in average values and standard deviation. The results were analyzed using a one-way analysis of variance (ANOVA) *p* ≤ 0.05.

### 2.5. Antibiofilm Activity

The following compounds were under investigation: HABH (10^−4^ M/mL), CuHABH, (10^−4^ M/mL), Hesperetin (10^−4^ M/mL) and DMSO. The concentration of 3 × 10^−4^ M/mL was used for all tested compounds. The growth and adhesion abilities of the bacterial strains in the presence of the tested compounds was carried out following previously published methods [45]. For the bacterial cultivation, the minimal medium of 50-fold diluted buffered peptone water (Merck, Darmstadt, Germany) was used because, according to the literature, biofilm formation may be stimulated in environments poor in carbon sources [46]. In this culture medium, the availability of the organic compounds was very limited and equal 0.1 g/L [45]. The sterile minimal medium was poured into sterile 25 mL Erlenmeyer flasks, into which sterile glass carriers (Star Frost 76 × 26 mm, Knittel Glass, Braunschweig, Germany) were placed vertically in such a way that half of the carrier was immersed in the medium while the other part remained outside. The amount of bacterial inoculum was standardized densitometrically in McFarland degrees [1 °McF] to obtain a cell concentration in the culture medium approximately equal to 5 × 10^3^–1 × 10^4^ CFU/mL at the start of each experiment. The samples were incubated at a temperature of 25 °C on a laboratory shaker (135 rpm) for 6 days. Growth of the bacterial cultures was evaluated in °McF using a densitometer DEN-1 (Merck, Darmstadt, Germany). In turn, analysis of the bacterial adhesion to the glass carriers was performed using luminometry. For the luminometric tests, the carrier was removed from the culture medium, rinsed with sterile distilled water and swabbed using free ATP sampling pens (Merck, Darmstadt, Germany). The measurements were reported in RLU/cm^2^ using a HY-LiTE 2 luminometer (Merck, Darmstadt, Germany). These experiments were conducted in triplicate, and the standard deviation was calculated. In addition, the bacterial attachment was observed using fluorescence microscopy (Nikon) after DAPI staining.

## 3. Results and Discussion

### 3.1. Characterization of the Complex CuHABH

#### 3.1.1. IR Spectral Studies

IR spectra provided substantial and valuable information on the coordination reaction. All the spectra recorded were characterized by vibrational bands mainly due to the NH, O–H, C=O and C=N groups (Appendix A). The IR spectrum of the Schiff base ligand HABH (LH_3_) showed a broad band at 3421 cm^−1^ due to phenolic OH and medium intensity weak bands at 3333 cm^−1^ due to—amide NH. The strong bands of high intensity observed at 1630 cm^−1^, 1609 cm^−1^ and 1253 cm^−1^ were due to the carbonyl function ν(C=O), azomethine function ν(C=N) and phenolic function(C–O), respectively. The IR spectra of the metal complex exhibited ligand bands with the appropriate shifts due to a complex formation. It was observed in the CuLH_2_(OAc) spectrum an absence of stretching vibration due to phenolic OH at 3421 cm^−1^. This may indicate the formation of a coordination bond between the metal ion and phenolic oxygen atom at C5 (Appendix A) via deprotonation. This was further confirmed by the increase of absorption frequency of phenolic ν(C–O) of about 25 cm^−1^ which appeared in the region 1278 cm^−1^ of the complex, indicating the participation of an oxygen atom of the phenolic OH in the coordination. The band assigned to the OH group did not completely disappear, and it was probably merged with the NH band.

In the spectrum of the metal complex a medium intensity weak band of ν(NH) at 3395 cm^−1^ was shifted to higher frequencies Δν = 62 cm^−1^ in comparison to a small and sharp vibration in the ligand. The amine group did not take part in the formation of the coordination bond with the copper, which was confirmed in EPR studies. The change of the band shape of ν(NH) in the complex was probably due to the merging of the OH and NH bands.

The position of the amide carbonyl frequency ν(C=O) in the spectrum of the copper complex observed at 1613 cm^−1^ was shifted to a lower frequency by approximately 17 cm^−1^ compared to that of the ligand. This indicates the participation of oxygen from the CO group of the amide in the coordination of copper ions [47]. The stretching frequency of the C=N group in the copper(II) complex observed at 1593 cm^−1^ was shifted to a lower wavenumber compared to that of the ligand observed at 1609 cm^−1^.

The results clearly indicate the coordination of a copper ion through the N atom of the azomethine group of the HABH Schiff base [48]. The presence of acetate anions in the complex was confirmed by the bands of stretching vibrations ν (COO-) at the frequencies of 1563 and 1362 cm^−1^. Coordination of the metal ion with the ligand was additionally confirmed by the appearance in the spectrum of the complex of new, non-ligand bands of weak intensity in the regions of 445 cm^−1^ and 410 cm^−1^ assigned to the stretching vibrations ν(M–O) and ν(M–N), where M is a metal ion, respectively.

#### 3.1.2. EPR Spectral Studies in Solid State and Solution

The EPR spectra of the solid complex CuHABH (or [CuLH_2_(OAc)]) were recorded on the polycrystalline powder at 298 K (room temperature, RT) and 77 K (liquid nitrogen temperature, LNT) and after dissolving the compound in DMSO and DMF at 77 K. The spectra, reported in Figure 2, were countersigned by a tetragonal signal. The resolution improved with decreasing the temperature, and at LNT (liquid nitrogen temperature) the hyperfine coupling between the unpaired electron and the ^63,65^Cu nucleus clearly emerges (Figure 2b).

To improve the resolution, the solid compound was dissolved in an organic solvent and the spectra were recorded on the solution (Appendix A). In DMF and DMSO, the features due to the hyperfine coupling were well visible and allowed us to fully characterize the structure of the complex. The EPR spectrum was assigned to only one species with *g*_z_ = 2.240–2.241 and |*A*_z_| = 192.5–194.0 × 10^−4^ cm^−1^. The spin Hamiltonian parameters (Table 1) were extracted generating the experimental spectrum with WinEPR SimFonia software [25]. The value of *g*_z_ and *A*_z_ fell in the expected range for an equatorial coordination Cu(NO_3_) [49]. The order *g*_z_ >> *g*_x_~*g*_y_ > *g*_e_ indicates a tetragonal symmetry with a small rhombicity degree and a ground state based on the Cu-*d*_x2–y2_ orbital [50]. In correspondence with the first resonance of the parallel region, the superhyperfine coupling with one nucleus of ^14^N (*A*^N^) with *I* = 1 was revealed (region amplified in Appendix A). The quartet detected showed an experimental intensity ratio of 1:3:3:2 which can be explained with the overlap of the two triplets with an approximate ratio of 1:1:1 and 2:2:2 due to the presence in the natural copper of the isotopes ^63^Cu (69.2% natural abundance) and ^65^Cu (30.8%). A detailed explanation of the origin of this multiplet is given in ref. [17]. The value of 16.0 × 10^−4^ cm^−1^ for *A*^N^ was in good agreement with those reported in the literature up till now [51,52,53,54]. Summarizing these findings and the data of the elemental analysis, the results suggest that a tetra-coordinate complex with square planar geometry is formed in the solid state with two oxygen and one nitrogen donors of the ligand occupying three equatorial positions, while an oxygen of an acetate ion completes the equatorial plane of the copper(II) ion.

The structure of the [CuLH_2_(OAc)] species was optimized by DFT methods using the B3LYP functional coupled with the Grimme’s D3 dispersion, and the basis set 6-311g(d,p) for the main group elements and SDD plus *f*-functions and pseudo-potential for the Cu, following the procedure in the literature [37]. The structure is shown in Figure 3.

The Cu–O distances were between 1.830 Å and 1.926 Å, while the length of the Cu–N was significantly larger (2.119 Å). The structure showed a slight distortion from the regular square planar geometry with *cis* and *trans* angles in the range of 82.3–96.4° and 168.5–177.8°, respectively.

In addition, the EPR spectrum was calculated by the DFT protocol recently validated on fourteen Cu^II^ complexes and, in particular, it was demonstrated that the mean absolute percent deviation (MAPD) in the prediction of *A*_z_ with B3LYP functional was 8.6% with a standard deviation (SD) of 4.2%, while the MAPD for the *g*_z_ with PBE0 was 2.9% with a SD of 1.1% [37]. The results are listed in Table 1: the *g*_z_ value is slightly underestimated with a percent deviation (PD) from the experimental data from −1.0% to −1.6%, while the *A*_z_ is overestimated with a PD in the range of 3.6–6.4%. These predictions are in line with the previous results in the literature [37,55]. Anisotropic EPR spectra were recorded in an aqueous solution containing ^63^CuSO_4_·5H_2_O and HABH (H_3_L) with a molar ratio of 1:1 and Cu(II) concentration of 1.0 × 10^–3^ M (Figure 3). After the formation of aquation [Cu(H_2_O)_6_]^2+^, the complexation process started around pH 4.0–4.5 and two major species were detected in the solution, indicated with I and II in Figure 4. The spin Hamiltonian parameters for I were *g*_z_ = 2.373 and *A*_z_ = 150.6 × 10^−^^4^ cm^−^^1^, that differed little from those of the [Cu(H_2_O)_6_]^2+^ (aq in Figure 4, *g*_z_ = 2.411 and *A*_z_ = 135.3 × 10^−^^4^ cm^−^^1^). A comparison with *g*_z_ and *A*_z_ for the (NH_2_, CO) coordination (2.313–2.328 and 153–164 × 10^−^^4^ cm^−^^1^ [56]) indicated that the donor set for I could be (imine-N, CO). The species II, formed above pH 7, was characterized by *g*_z_ = 2.245 and *A*_z_ = 188.8 × 10^−^^4^ cm^−^^1^, comparable to the parameters measured for the solid compound dissolved in the DMSO and DMF (Table 1). These values were also similar to the *g*_z_ and *A*_z_ detected for the hesperetin Schiff base, which had the same equatorial coordination [17]. 

The comparison of the spectra recorded on the solid compound dissolved in DMSO and on the aqueous solution containing ^63^CuSO_4_·5H_2_O and HABH (in the form of LH_3_) with a molar ratio of 1:1 at pH 8.05 suggests that the two species share the same coordination mode (Appendix A). This implies that in the [CuHABH] species, revealed by potentiometry in the pH range 6.5–9.5, the ligand binds Cu(II) in the tridentate form with the donor set (O^−^, imine-N, CO), similar to that observed in the solid state for the [CuLH_2_]^+^ (Figure 3). The deprotonation of the last non-coordinating 3’-OH group would give the species [CuL]^–^, as suggested by potentiometry (Appendix A), with the same coordination mode and, consequently, the same spin Hamiltonian parameters. The spectra obtained using the isotope ^63^Cu, moreover, allowed us to confirm the binding of only one nitrogen donor to the copper. In fact, the first parallel resonance at the lowest magnetic field showed only three resonances with the intensity ratio 1:1:1 expected for the coupling of the unpaired electron with one ^14^N nucleus. The values for *A*^N^ was 15.6 × 10^−4^ cm^−1^, comparable with those measured upon dissolution of the solid CuHABH in an organic solvent and with those reported in the literature [51,52,53,54].

### 3.2. Interaction of the Compounds with CT DNA

The binding affinity of the complexes with DNA may provide the basis for a number of potential biomedical applications targeting the destruction of bacteria. The interaction of HESP, HABH and CuHABH with CT DNA was investigated by UV-Vis spectroscopy, via competitive studies with TO monitored by fluorescence emission spectroscopy and CD spectroscopy. Electronic absorption spectroscopy was a preliminary and important technique to investigate the binding mode and the strength of the compounds with the CT DNA, through titration and the calculation of the DNA-binding constant (*K_b_*) [57]. The CT DNA binding studies concerned the recording of the UV spectra of the CT DNA in the presence of increasing amounts of compounds (for different r = [complex]/[CT DNA]). The binding was usually accompanied by noticeable changes in UV–is absorbance. During the titration, any change(s) in the band of CT DNA at 258–260 nm, in the charge transfer or intraligand band(s) of the compounds, may indicate the existence of some kind of interaction [58]. UV-Vis titrations of all the three compounds with CT-DNA were performed in a Tris-HCl(NaCl) buffer (pH 7.2). 

The absorption profiles of the HESP and HABH showed two absorption bands λ_max_ = 288, 323 nm and λ_max_ = 285, 330 nm, respectively, and the CuHABH complex exhibited one absorption band at λ_max_ = 371 nm (Appendix A). The bands at the longer wavelength may be attributed to intraligand or metal-induced charge transfer bands. Generally, the transfer bands are environmentally sensitive so, it was expected that these bands would show substantial changes in the presence of biological and bio-mimicking diverse microenvironments. It was found that the HESP showed a hyperchromic shift whereas the HABH and CuHABH showed a hypochromic shift with increasing concentration of the CT-DNA (Appendix A, Table 2). Slight hyperchromism in the absorption band of the HESP at 323 nm may indicate DNA minor groove interactions [59,60]. On the other hand, significant hypochromism of the HABH and CuHABH (Appendix A, Table 2) in the presence of double helical DNA was characteristic of the interactions between DNA and the molecule, which was due to the strong stacking interaction between the aromatic chromophore and the base pairs. Hence, it can be considered that there were some interactions between the HABH, CuHABH and the DNA.

The coupling π orbital was partially filled by electrons, thus decreasing the transition probabilities and concurrently resulting in hypochromism [61]. On the other hand, according to the literature reports, hypochromism indicates a strong interaction between the electronic states of the chromophore and that of DNA bases. As the decrease in the strength of the electronic interaction was expected as the cube of the distance between the chromophore and the DNA bases [62], the observed large hypochromism in our experiment strongly suggests a close adjacency of the HABH and especially the CuHABH to the DNA bases. 

The absence of an isosbestic point in the absorption spectra was indicative of more than one type of binding interaction between the complexes and the CT-DNA. The binding constants were calculated for the interaction of the compounds with the CT-DNA using the Wolfe–Shimer equation (Equation (1)). From the plots of [*DNA*]/(*ε_a_* − *ε_f_*) vs. [*DNA*] (Appendix A, inset), linear relationships were acquired. The intrinsic binding constants (*K_b_*) were calculated from the ratio of the slope and intercept. They showed the following trends: *K_b_*^CuHABH^ > *K_b_*^HABH^ > *K_b_*^HESP^. The values were in conformity with the observed trend in hypochromism (Table 2). This result indicates that the extent of interaction of the HABH or CuHABH was much greater than that of the HESP, and in turn the CuHABH exhibited a greater effect compared to the HABH suggesting its higher propensity for DNA binding. The high binding constant (*K_b_*) for the complex may be attributed to steric hindrance imposed by the coordinated ligand. Lower values also indicated partial intercalative binding and/or surface binding of the compounds to the DNA.

To further clarify the binding mode of the complexes with the CT-DNA, TO competitive studies were carried out by fluorescence emission spectroscopy to examine whether the compounds tested can replace TO in the TO bounded CT-DNA system. Thiazole orange is composed of two heterocyclic ring systems (quinoline and benzothiazole) connected through a methine bridge (Figure 5).

The fluorescence intensity of the TO depends upon its conformation [63]. A planar state allows conjugation between the two aromatic systems—this is the fluorescent form—whereas rotation at the methine bridge produces a non-planar conformation, which is not fluorescent. In the presence of double stranded (ds) DNA, the TO acts as an intercalator (or groove binder) [64,65]. When intercalated, its fluorescent planar conformation is stabilized by stacking between base pairs [66]. The intercalating effect of the compounds was studied by adding stepwise a certain amount of their solution into a solution of the DNA-TO adduct. The impact of the addition of each compound to the DNA-TO adduct solution was obtained by monitoring the changes of the fluorescence emission spectra recorded with the excitation wavelength (λ_ex_) at 509 nm (Appendix A). The addition of the compounds induced moderate quenching of the fluorescence emission band of the DNA-TO adduct with respect to the HESP, or significant with respect to the HABH (70%) and CuHABH (90%) (Appendix A). The Stern–Volmer (*K_SV_*) and the quenching constants (*k_q_*) were determined using the Equations (2) and (3) (presented in Section 2). In turn, the values of the association constants (*K_a_*) and the number of binding sites (*n*) were calculated on the base of the logarithmic Equation (4):(4)$$\log\left(I_{0}-I/I_{0}\right)=logK_{a}+nlog\left[Q\right]$$
where *I*_0_ and *I* are fluorescence intensities of DNA-TO adduct in the absence and presence of the tested compounds, respectively, and [*Q*] compound concentration. The results are reported in Table 3.

The number of binding sites ranged from 0.55 for the HESP, 1.01 for the HABH to 1.63 for the CuHABH indicating the presence of a single binding site in the CT-DNA-TO adduct. The quenching constants were significantly higher than 10^10^ M^−1^ s^−1^ suggesting that the quenching of the DNA-TO fluorescence induced from the compounds possibly took place via a static mechanism leading to the formation of a new adduct, obviously between the DNA and the HESP, HABH or CuHABH. This may confirm indirectly the TO-displacement and subsequently partial intercalation of the compounds to the CT DNA. Further, the apparent binding constant (*K_app_*) values were found for the compounds using the equation *K_app_* = *K*_TO_ × [C_TO_]/[C_50_] (Table 3). The data suggest that the interaction of the CuHABH with CT-DNA was the highest one as compared to the HABH and HESP, which is consistent with the above absorption and emission spectral observations. 

Since these changes indicate only one kind of quenching process, it may be concluded that all of the compounds bind to CT-DNA via a similar mode. Furthermore, such quenching constants and apparent binding constants of the ligand and Cu(II) complex (order of magnitude 10^6^) suggest that the interaction of the compounds with DNA should be mainly intercalation although other pathways of interaction (electrostatic, minor/major groove) cannot be excluded.

Circular dichroism spectroscopy is a useful tool for conformational investigation of the DNA helix. For this purpose, CD spectra of CT-DNA in the presence and absence of the HABH and CuHABH were recorded. These compounds were selected because of their stronger activity confirmed by electron absorption and emission spectroscopy. The CD spectrum of the bare DNA showed a positive band at 276 nm and a negative one at 242 nm which are the characteristic bands of the B-form of CT-DNA. In the case of groove binding or electrostatic binding, the CD spectrum showed almost no change, whereas the intercalative mode of binding affected both the positive and negative bands. With an increasing concentration of the HABH, the ellipticity of the positive band increased and that of the negative band decreased to a large extent (Figure 6).

In the positive band a red-shift (2 nm) was observed. In the case of the complex CuHABH, the extent of change was the same in a positive band and opposite in a negative band (intensity increasing) in comparison to that of the HABH, but in a narrow concentration range (10–50 μM). Above the concentration of 75 μM new bands appeared at 367 and 415 nm. Their occurrence was most likely related to the formation of the CT-DNA-CuHABH adduct, which may have led to a drastic change in the spectrum of the CuHABH complex. The results indicate a stronger interaction between the CuHABH than the HABH with the CT-DNA. It can be assumed that the nature of the interactions was of the intercalating type, but in the high concentration range, the complex could bind DNA externally to phosphate residues or to a minor/major groove forming an adduct.

### 3.3. Antibacterial Activity of the HABH and CuHABH Complexes

In vitro antibacterial screening of the ligand (LH_3_) and the metal complex [CuLH_2_(OAc)] was assayed against various bacterial strains and the potentiality of the compounds was evaluated by measuring the diameter of the zone of inhibition in mm. The report of the antimicrobial results presented in Table 4 revealed that both the free ligand and its metal complex exhibited variable antimicrobial activity depending on the type of bacterial strain.

However, their activity is lower than that of standard antibiotics. The tested compounds showed variable activity against three Gram-positive bacteria (*Staphylococcus aureus* ATCC 25923, *Staphylococcus aureus* ATCC 27734 and *Listeria monocytogenes* ATCC 19115 isolation-human). The original hesperetin revealed the lowest inhibitory effect towards both strains of *Staphylococcus aureus* but the highest to *Listeria monocytogenes* ATCC 19115 as compared to the HABH and CuHABH (statistically significant). On the other hand, all Gram-negative bacteria (*Salmonella* Choleraesuis, *Salmonella* Typhimurium, and *Escherichia coli*) and one Gram-positive *Listeria monocytogenes* ATCC 19111 (isolation poultry) seem to be resistant to both ligands (HABH, HESP) and the complex (CuHABH) under the conditions of the measurements. The differences in the sensitivity to antagonistic factors of bacteria belonging to the same species may be manifold, e.g., the source of origin or differences consisting in different antigenic determinants located on the cell surface. Antigenic differences result from the presence of various chemical compounds, such as membrane proteins or lipoteichoic acid, which determine the structure of the cell membrane [67]. Considering the literature data, it can be assumed that there were a number of factors possibly affecting the response of the bacteria to the compounds tested in this study, including, diverse detoxification systems dependent on the bacteria strain or culture conditions that influenced the compounds’ toxicity. These may change the cell surface composition of bacteria and thus reduce their viability [68].

In the case of Gram-negative bacteria used in these studies, no antagonistic activity of the tested compounds was found. This phenomenon may have resulted from the ability of the Gram-negative bacteria to create so-called outer membrane vesicles (OMV). OMVs play a special role in the physiology of Gram-negative bacteria. The creation of OMVs results in the encapsulation of stinging agents such as toxic substances inside the cell, preventing them from entering the cell. Thus, *E. coli* mutants showing very high levels of OMVs secretion have been shown to be resistant to polymyxin B compared to the wild-type strain [69]. Another example is the ability of the *Pseusomonas aeruginosa* to break down β-lactam antibiotics. *P. aeruginosa* strains in the presence of these antibiotics produce OMVs that contain enzymes that break down and inactivate β-lactam [70]; however, in order to confirm the above mechanism in our case, additional research is necessary.

There are many mechanisms of resistance to chemotherapeutic agents, but they are based on several major strategies. The most common are the production of enzymes that inactivate antibiotics, a blockade of antibiotic target receptors, changes in the permeability of the cell membrane and the mechanism of active removal of the antibiotic from the cell (efflux pump). The efflux pump mechanism is based on pumping out the chemotherapeutic agent, e.g., antibiotics, out of the cell [71]. The available literature describes studies on blocking the expression of efflux pomp proteins by bioactive compounds. The research group of Tintino et al. [72] demonstrated the inhibitory effect of tannic acid against efflux pumps expressed by the *Staphylococcus aureus* RN4220 and IS-58 strains. The efflux pump inhibition was assayed using a subinhibitory concentration of efflux pump standard inhibitors and tannic acid (MIC/8), observing their capacity to decrease the MIC of ethidium bromide and antibiotics due the possible inhibitory effect of these substances. In other studies, the effectiveness of inhibiting the efflux pump mechanism was also demonstrated in *Staphylococcus aureus* strains by caffeic and gallic acid (the MrsA pumps of the strain RN-4220 and NorA of the strain 1199B) [73], hydroxyamines derived from lapachol and norlachol (NorA pumps from the strain 1199B) [74], ferulic acid and its esterified derivatives (NorA pumps from the strain 1199B) [75]. In contrast, studies [76] have shown the activity of blocking the efflux pump by quercetin. Quercetin showed synergism with all antibiotics tested. In the association of this substance with ethidium bromide, there was a reduction of MICs in the strains carrying the TetK pump (from 16 to 2.51 μg/mL) and NorA (from 64 to 12.69 μg/mL), which shows the inhibition of these mechanisms.

### 3.4. Antibiofilm Activity

Bacterial adhesion to solid surfaces is a general phenomenon with relevance to numerous medical, industrial and ecological problems. Adhesion to inert materials, such as plastic and glass, has been recognized for several years as resulting from the physicochemical interactions between different components (microorganism, substrate and suspending medium) [77]. According to the XDLVO (Derjaguin, Landau, Verwey, and Overbeck) theory, the microbial adhesion is described as a balance between the Van der Waals, the electrostatic and the Lewis acid–base interactions (Figure 7) [78].

The magnitude of these interactions is affected by the distance of the bacterium from the surface and the ionic strength of the surrounding environment, but it is also necessary to take into account the structure of the bacterial cell wall which mediates the adhesion processes [79,80]. Therefore, in our studies, we used both Gram-positive and Gram-negative bacteria as the model microorganisms. Glass was selected for the experiments as a model surface because it is widely studied and most of its physicochemical properties are known and available in the literature [81]. Adhesion of bacterial cells to the glass surface is presented in Figure 8.

The highest levels of RLU corresponding to organic matter and ATP were recorded for the DMSO solution. The best antiadhesive action was detected for the CuHABH complex for both bacterial strains. This was confirmed in microscopic observations using fluorescence microscopy after DAPI staining [82] (Figure 9).

All tested compounds in the concentration 10^−4^ M/mL had antimicrobial and antibiofilm activity against both Gram-negative and Gram-positive bacteria. In inoculated and cultivated samples with the HABH, CuHABH and HESP, growth of the bacterial strains was not detected densitometrically [0 °McF]. Only in the culture media with the DMSO was very weak bacterial growth (very low turbidity) measured [0.3 °McF]. Interestingly, the best action against biofilm formation was detected for the Cu-HABH. Metal-based nanoparticles are known from non-specific bacterial toxicity [83]. They do not bind to a specific receptor in the bacterial cell which not only makes the development of resistance by bacteria difficult, but also broadens the spectrum of antibacterial activity. Both Gram-positive and Gram-negative bacteria have a negatively charged surface, however, there are significant differences in the structure of the cell wall [84]. Gram-positive bacteria have a thick layer of peptidoglycan formed by linear chains of N-acetylglucosamine and N-acetylmuramic acid. On the other hand, Gram-negative bacteria, have a more complex structure with a thin layer of peptidoglycan and a phospholipid outer membrane with partially phosphorylated lipopolysaccharides contributing to increase the negative surface charge of a cell envelope. It has been documented that positively charged nanoparticles establish a strong bond with negatively charged bacterial membranes, resulting in the disruption of cell walls and increasing their permeability. In addition, nanoparticles can also release compounds from the extracellular space, capable of entering the cell and disrupting biological processes. Metal-based nanoparticles, due to their non-specific mechanisms of antimicrobial action, generally exhibit a broad spectrum activity [83,85].

Among the metal particles, copper compounds have been explored with great attention as antimicrobial and antibiofilm agents [86,87]. The effects of copper are multifaceted and include, for example, the destruction of iron–sulfur cluster proteins and metalloproteins, production of reactive oxygen species, and interference with membrane integrity. These mechanisms are possibly the key to the antibacterial action of this metal. It has been documented that copper compounds are proven to restrict the spread of epidemic methicillin-resistant *S. aureus* (MRSA) strains. Therefore, the nonselective nature of these copper-dependent processes poses a challenge for target-directed applications in antibacterial strategies [88]. The copper compounds also exhibit antibiofilm activity with the ability to disrupt pre-formed biofilms. In studies conducted by Brahma et al., hexadentated macrocyclic complex of copper(II) showed an increased susceptibility towards *S. aureus* and it was able to reduce more than 95% of the bacterial load at the concentration10 μg/mL [89]. The strong inhibitory effect of CuHABH was also observed in our studies.

Furthermore, the presence of a flavonoid molecule may also affect the biofilm formation and quorum sensing phenomenon, but this mechanism is not fully understood [9]. It was documented that the administration of flavonoids to *P. aeruginosa* cultures alters the transcription of quorum sensing-controlled target promoters and suppresses virulence factor production. Recent researches show that flavonoids are able to inhibit biofilm formation by *Streptococcus mutans* [90], *Aeromonas hydrophila* [91], *Candida albicans* [92], *S. aureus* [93], and *Escherichia coli* O157:H7 [94] and persister cells formation in *A. baumannii* [95]. Flavonoids such as quercetin have the most potent antibiofilm property as they have the ability to inhibit DNA gyrase, bacterial energy metabolism and cell membrane function [96]. It has also been shown that aglycone flavonoids, including hesperetin, inhibit biofilm formation by *S. aureus* strains that overexpress efflux protein genes. These effects are more strongly established by aglycone flavonoids than their glycone [97]. This phenomenon was confirmed in our study. The hesperetin molecule exhibited more potent antibiofilm property than its hydrazone (modified molecule of hesperetin) in terms of the bacteria used as shown in Figure 8.

Summarizing, the CuHABH complex may play a crucial role in disrupting exopolymers in biofilm formation. Nevertheless, more research is needed to determine the mechanism of the action of CuHABH as a potential biocide.

## 4. Conclusions

In summary, the new derivative of hesperetin, HABH (2-amino-N′-(2,3-dihydro-5,7-dihydroxy-2-(3-hydroxy-4-methoxyphenyl)chromen-4-ylidene)benzohydrazide), and its copper complex, CuHABH, were designed in order to investigate their antimicrobial and antibiofilm activities. Their structures both in a solid state and in solution were established using several methods: FT-IR, ^1^H NMR, ^13^C NMR, UV-Vis, FAB MS, EPR, ESI-MS and potentiometry. In the copper complex dominating at the physiological pH region, to which the potentiometry assigned the stoichiometry CuLH species, the ligand binds to Cu(II) in the tridentate form, with the donor set (O^−^, imine-N, CO), analogous to that observed in the solid state for [CuLH_2_]^+^. Furthermore, the interaction between HABH and CuHABH with calf-thymus DNA was investigated. The results indicate a stronger affinity of the CuHABH to the CT-DNA than the HABH. It can be assumed that if the copper complex penetrates the outer membrane of bacterial cells, it can disrupt/stop DNA or protein synthesis, which will result in limiting the growth of bacterial cells, enhancing the anti-adhesive effect against bacterial strains.

Hesperetin and its derivatives show a wide range of antimicrobial activity [9]. Antibacterial screening of the ligand and the metal complex was assayed against Gram-positive bacteria: *Staphylococcus aureus* ATCC25923, *Staphylococcus aureus* ATCC27734, *Listeria monocytogenes* ATCC 19111, and *Listeria monocytogenes* ATCC 19115 and Gram-negative bacteria: *Salmonella* Choleraesuis ATCC 7001, *Salmonella* Typhimurium ATCC 14028, and *Escherichia coli* ATCC 10536, and the potentiality of the compounds was evaluated by measuring the diameter of the zone of inhibition. The antagonistic activity of natural and synthetic bioactive compounds depends on the factor-strain test system. Very often these are individual interactions, which we observed within the tested Listeria monocytogenes strains; however, in the case of the strains of Gram-negative bacteria and the tested HABH or CuHABH, bacterial resistance to these compounds was observed. Under the experimental conditions of this study, it is concluded that both the geometry and Cu presence may play an important role in antibiofilm activity. Among different compounds tested in this study, the CuHABH was the most effective antimicrobial material against both Gram-positive and Gram-negative bacteria. It diminished the adhesive action of bacteria. This molecule may play a crucial role in disrupting exopolymers (DNA/proteins) in biofilm formation. The CuHABH complex could be used to prevent bacterial adhesion especially on glass equipment. The present study had its own limitations as only the antibiofilm efficacy for two reference bacterial strains was assessed. The mechanism of interaction requires in-depth research to elucidate the phenomenon. Further studies are required to assess the impact of these compounds on wild bacterial strains isolated from various natural environments. It is evident from this work that the studied compounds, especially the CuHABH complex, may be efficient alternatives in the search for new forms of treatment against bacteria. The knowledge gained from the conducted research will be used in future studies, concerning the formation of nanocapsules of flavonoid derivatives with increased water solubility.

Our research should be considered significant; however, more research is needed to investigate the detailed antimicrobial and anti-biofilm effects of hesperetin and its derivatives. This compound, as well as other flavonoid structures, will always inspire research into the design and synthesis of new effective antimicrobials.

## Figures and Tables

**Figure 1 molecules-27-00845-f001:**
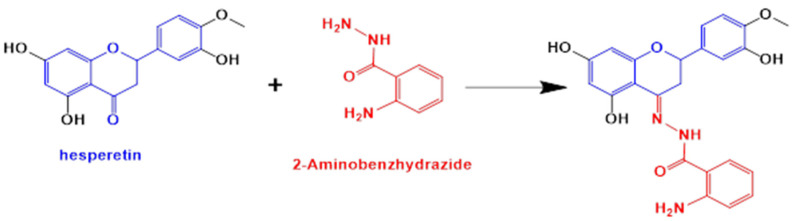
Synthesis of 2-amino-N′-(2,3-dihydro-5,7-dihydroxy-2-(3-hydroxy-4-methoxyphenyl)chromen-4-ylidene)benzohydrazide (HABH).

**Figure 2 molecules-27-00845-f002:**
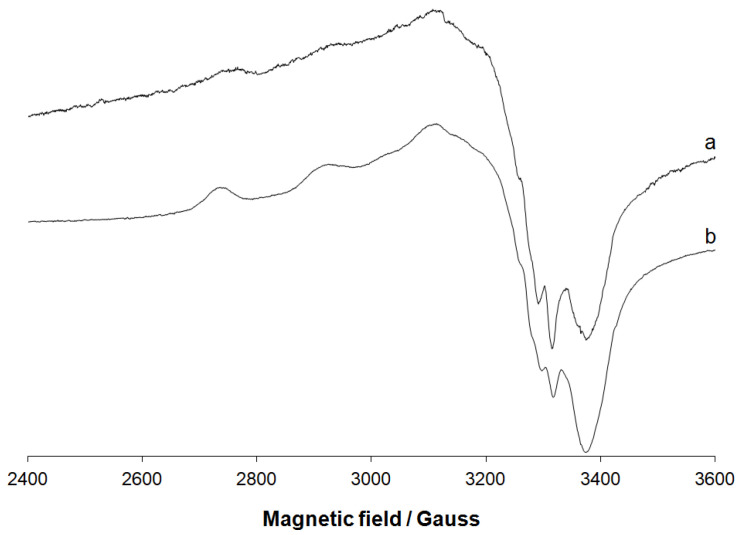
EPR spectra recorded on the polycrystalline powder of [CuLH_2_(OAc)]: (**a**) 298 K and (**b**) 77 K.

**Figure 3 molecules-27-00845-f003:**
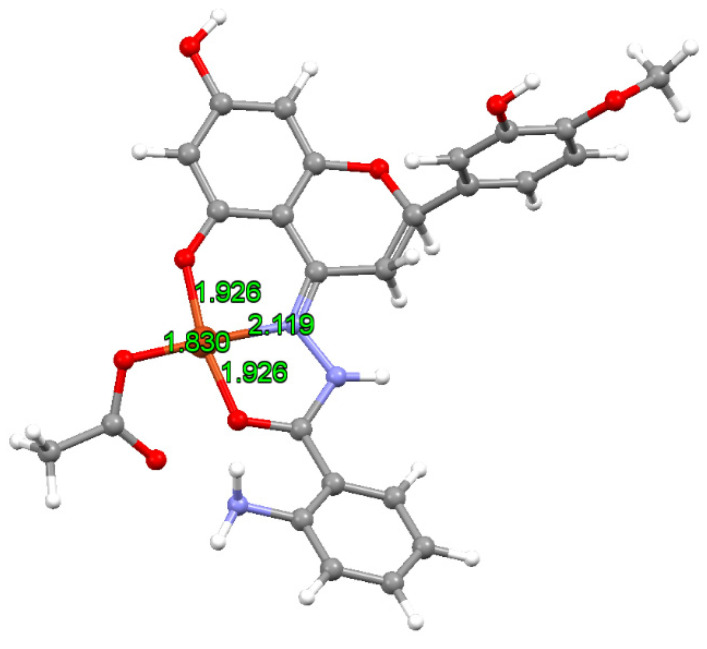
Optimized structure of [CuLH_2_(OAc)]. The distances of Cu–donors are also shown (in Å).

**Figure 4 molecules-27-00845-f004:**
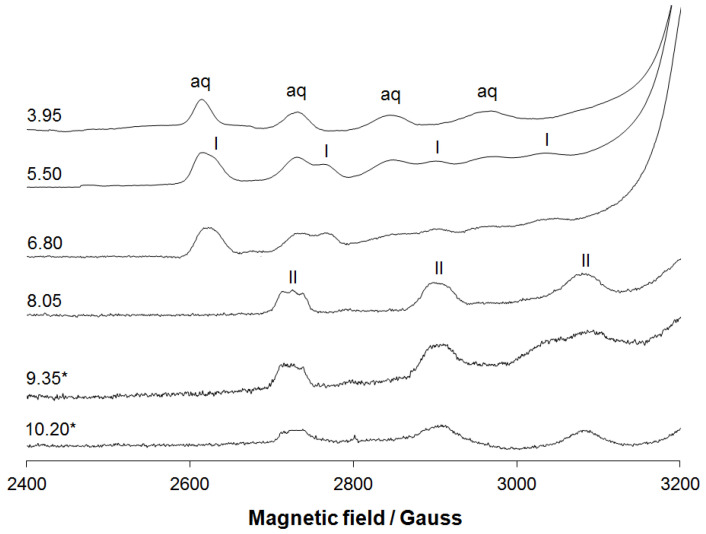
Low field region of the X-band anisotropic EPR spectra recorded at 77 K as a function of pH in an aqueous solution containing ^63^CuSO_4_·5H_2_O and HABH (LH_3_) with a molar ratio of 1:1 and Cu(II) concentration of 1.0 × 10^–3^ M. The values of pH are shown for each spectrum. The asterisk indicates that at pH 9.35 and 10.20 the solution was opalescent for the presence of the solid compound. With aq, I and II the first parallel resonances of aquation, and of the complexes with coordination (imine-N, CO) and (O^−^, imine-N, CO) are denoted.

**Figure 5 molecules-27-00845-f005:**
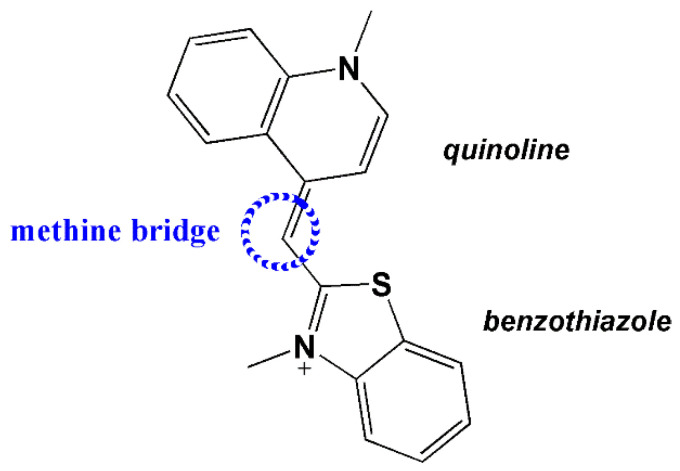
Structure of thiazole orange (TO).

**Figure 6 molecules-27-00845-f006:**
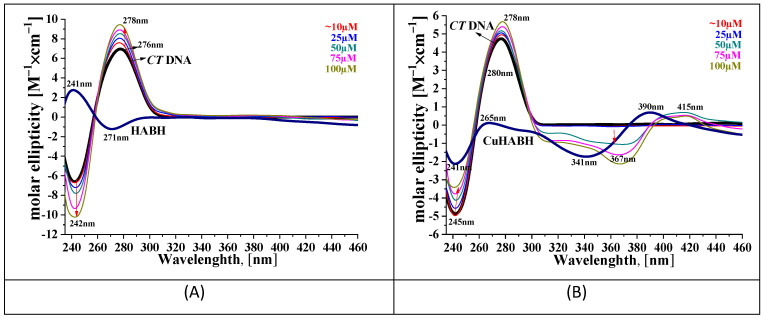
CD spectra of CT-DNA (100 µM) in the presence of: (**A**) HABH, and (**B**) CuHABH at 0–100 µM concentration. The red arrows indicate the direction of change/movement of the spectrum. The black arrows show the CT-DNA spectrum of DNA and its absorbance maximum. The numbers shown on the spectra refer to the absorbance maximum of the emerging bands.

**Figure 7 molecules-27-00845-f007:**
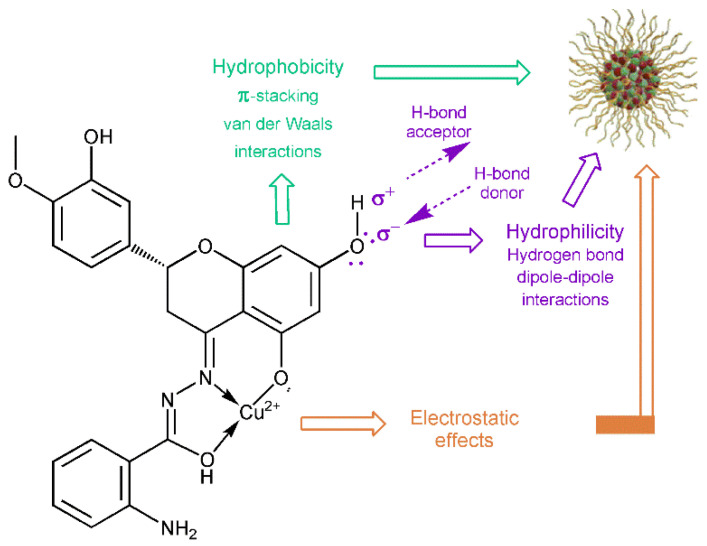
Proposed interactions of CuHABH complex with bacteria.

**Figure 8 molecules-27-00845-f008:**
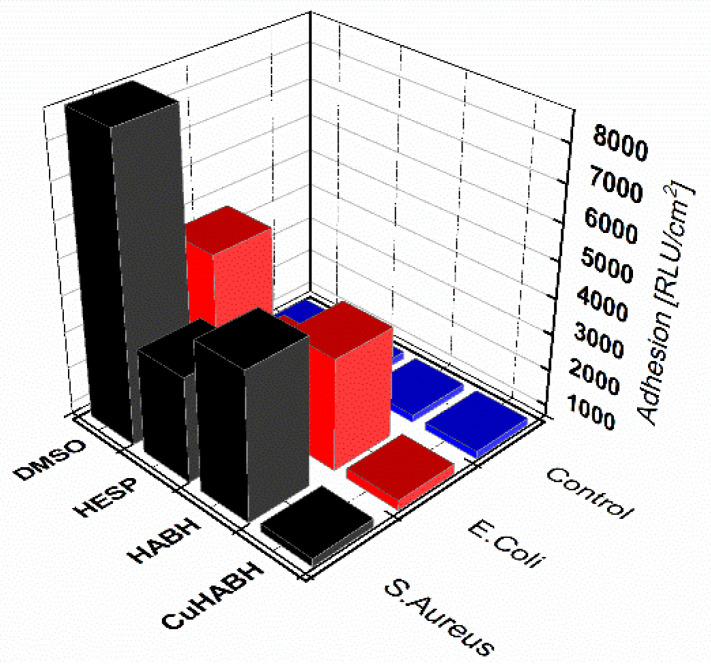
Adhesion [RLU/cm^2^] of bacterial strains to glass surface after 6-day incubation. DMSO (solvent); HESP (aglycone flavonoid); HABH (hesperetin hydrazone); CuHABH (complex Cu(II) with HABH).

**Figure 9 molecules-27-00845-f009:**
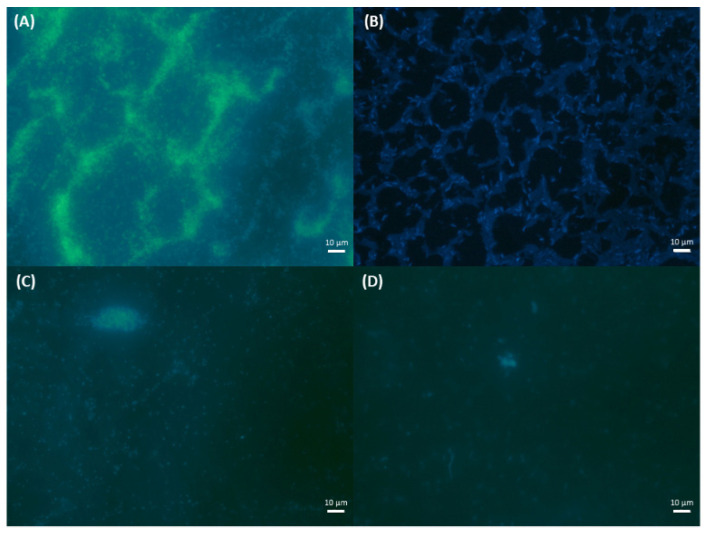
Adhesion of bacterial cells to glass surface after 6-day incubation and DAPI staining. (**A**) *S. aureus* and DMSO; (**B**) *E.coli* and DMSO; (**C**) *S. aureus* and CuHABH, and (**D**) *E. coli* and CuHABH. Bars represent 10 μm.

**Table 1 molecules-27-00845-t001:** Experimental and calculated spin Hamiltonian parameters of [CuLH_2_(OAc)].

Species	*g* _x_	*g* _y_	*g* _z_	PD(*g*_z_) ^a^	*A*_x_(Cu) ^b^	*A*_x_(Cu) ^b^	*A*_x_(Cu) ^b^	PD(*A*_z_) ^a^
Powder	c	c	2.232	−1.0	c	c	−193.4	3.9
DMF	2.040	2.060	2.240	−1.4	−11.0	−14.0	−192.5	4.4
DMSO	2.038	2.061	2.241	−1.4	−12.0	−13.0	−194.0	3.6
Aqueous solution ^d^	2.039	2.061	2.245	−1.6	−11.5	−13.0	−188.8	6.4
DFT calculated ^e^	2.061	2.062	2.209	–	−7.7	−18.5	−200.9	–

^a^ Percent deviation of the DFT calculated parameter from the experimental value. ^b^ Hyperfine coupling constant reported in 10^−4^ cm^−1^ units. c Parameter not measurable from the experimental spectrum. ^d^ Species detected at pH 8.05 with Cu(II)/HABH (H_3_L) ratio of 1:1 and Cu(II) concentration of 1.0 × 10^–3^ M. ^e^ The *g* and *A* tensors calculated by DFT methods at the level of theory PBE0/6–311g(d,p) (tensor *g*) and B3LYP/6–311g(d,p) (tensor *A*).

**Table 2 molecules-27-00845-t002:** UV–Vis spectral parameters of the interaction of HESP, HABH and CuHABH with CT DNA.

Compound	*K_b_* [M^−1^]	λ_max_ [nm]	(*A*_free_ − *A*_bound_)/*A*_free_ [%]_max_ (+)Hyperchromism/ (−)Hypochromism
HESP	1.20 (±0.15) × 10^4^	323	+5.57
HABH	4.00 (±0.22) × 10^5^	330	−41.78
CuHABH	5.00 (±0.18) × 10^5^	371	−56.52

**Table 3 molecules-27-00845-t003:** TO-DNA fluorescence quenching parameters: Stern–Volmer constant (*K_SV_*), apparent constant (*K_app_*), quenching constant (*K_q_*), number of DNA binding sites (*n*) and association constant (*K_a_*).

Compound	*K_SV_* [M^−1^]	C_50_ [μM]	*K_app_* [M^−1^] *	*K_q_* [M^−1^]	*n*	*K_a_* [M^−1^] **
Hesperetin	3.73 (±0.06) × 10^3^	nd ***	nd ***	1.43 (±0.20) × 10^12^	0.55 ± 0.04	1.58 (±0.20) × 10^3^
HABH	3.88 (±0.13) × 10^4^	40.4	1.86 × 10^6^	1.49 (±0.15) × 10^13^	1.01 (±0.03)	2.78 (±0.15) × 10^4^
CuHABH	1.31 (±0.06) × 10^5^	38.4	1.95 × 10^6^	5.04 (±0.14) × 10^13^	1.63 (±0.03)	6.74 (±0.14) × 10^4^

* *K_app_* = *K*_TO_ × [C_TO_]/[C_50_]_50% intensity decrease_; [C_TO_] = 25 µM, *K*_TO_ = 3.0 × 10^6^ [M^−1^]. ** log(*I*_0_ − *I*)/*I*_0_ = log *K_a_* + *n*log[*Q*]. *** nd—not determined in the range of tested concentrations.

**Table 4 molecules-27-00845-t004:** Antimicrobial activity of selected compounds against some bacteria strains as diameter zone of inhibition.

Compound 0.1 μM	Zone of inhibition [mm]						
	*Salmonella* Choleraesuis ATCC 7001	*Salmonella* Typhimurium ATCC 14028	*Escherichia coli* ATCC 10536	*Staphylococcus aureus* ATCC 25923	*Staphylococcus aureus* ATCC 27734	*Listeria monocytogenes* ATCC 19111	*Listeria monocytogenes* ATCC 19115
CuHABH	0.0 ± 0.0 ^aA^	0.0 ± 0.0 ^aA^	0.0 ± 0.0 ^aA^	10.5 ± 0.50 ^aA^	10.5 ± 0.50 ^aA^	0.0 ± 0.0 ^aA^	12.5 ± 0.50 ^aA^
HABH	0.0 ± 0.0 ^aA^	0.0 ± 0.0 ^aA^	0.0 ± 0.0 ^aA^	11.5 ± 0.50 ^aA^	12.0 ± 0.0 ^bA^	0.0 ± 0.0 ^aA^	13.5 ± 0.50 ^bA^
HESP	0.0 ± 0.0 ^aA^	0.0 ± 0.0 ^aA^	0.0 ± 0.0 ^aA^	9.0 ± 0.0 ^bA^	9.0 ± 0.50 ^cA^	0.0 ± 0.0 ^aA^	14.5 ± 0.50 ^cA^
DMSO	0.0 ± 0.0 ^aA^	0.0 ± 0.0 ^aA^	0.0 ± 0.0 ^aA^	0.0 ± 0.0 ^cA^	0.0 ± 0.0 ^dA^	0.0 ± 0.0 ^aA^	0.0 ± 0.0 ^dA^
Vancomycin	10.5 ± 0.00^B^	Nd	14.5 ± 0.50^B^	16.00 ± 0.00 ^B^	17.5 ± 0.00 ^B^	20.0 ± 0.80 ^B^	30.5 ± 0.50 ^B^
Ampicillin	Nd	12.60 ± 0.90 ^B^	Nd	Nd	Nd	21.0 ± 0.80 ^B^	16.5 ± 0.50 ^C^

Vancomycin (0.021 μM), Ampicillin (0.006 μM)–positive control. Nd–not determined. ^a,b,c,d^ statistical differences within one microorganism. ^A,B,C^ statistical differences within one microorganism in relation to the positive test.

## Data Availability

The data presented in this study are available on request from the corresponding author.

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
