# Peer review of "From the Physicochemical Characteristic of Novel Hesperetin Hydrazone to Its In Vitro Antimicrobial Aspects"

_molecules, 2022, doi:10.3390/molecules27030845_

Round 1
Reviewer 1 Report
In the manuscript entitled “From Physicochemical Characteristics of Novel Hesperetin Hydrazone to in Vitro Antimicrobial Aspects,” the authors was synthesized, analyzed the chemical structure by spectrometrical technical, procedure the physicochemical properties, and evaluate the antibacterial and antibiofilm biological activity. Overall, the work is well done, carefully thought, and performed, and the manuscript is well written and easy to read and follow. All experimental methods are well explained. The methods used are consistent with the literature and corroborate the objectives. The results presented are significant, robust and their presentation, interpretation, and conclusions are supported by other data present in the literature. Other Specific comments:
The authors inform that “Paper disks (∅ = 6 mm, Oxiod, 0Thermo Fisher Scientific, USA) were impregnated with compound samples, to obtain a concentration of test compounds of 0.1 μM per disk” this concentration information was confirmed by chemical analysis? This point is crucial to antibacterial activity.
The negative point of the work is the microbiological assay based on disc diffusion. The relevance of the antimicrobial assay used in this work (diffusion disc method) in the screening method based on the diffusion method is not recommended by the NCCLS for testing antimicrobial activity and is only used in the antibiotic susceptibility test. The antimicrobial screening test based on the disc diffusion method presents a series of experimental phenomena that can lead to false-negative results, such as difficulty in permeation of compounds in the agar medium, standardization of the disc concentration, dissolution of the essential oil, another problem that is not possible. determine the actual concentration of compounds in contact with the bacteria used in the test.
The purpose of the Interaction of the Compounds with CT DNA based only on UV–vis spectroscopy is very simplist. Others technical as RMN, Crystallography, or fluorescence emission spectroscopy are necessary to confirm the complex formation.
It is suggestive the calculate the IC50 for the antimicrobial activity. The MIC based only % of concentration no show the clinical relevance.
The results and, principally, discussion sections, not present effective indications of the mechanism of actions. Among so many articles, I suggest the following as an example:
Dos Santos, JFS, et al. "Enhancement of the antibiotic activity by quercetin against Staphylococcus aureus efflux pumps." Journal of Bioenergetics and Biomembranes 53.2 (2021): 157-167.
Pinheiro, PG, et al. "Antibacterial activity and inhibition against Staphylococcus aureus NorA efflux pump by ferulic acid and its esterified derivatives." Asian Pacific Journal of Tropical Biomedicine 11.9 (2021): 405.
Figueredo, FG, et al. "Effect of hydroxyamines derived from lapachol and norlachol against Staphylococcus aureus strains carrying the NorA efflux pump." Infection, Genetics and Evolution 84 (2020): 104370.
Tintino, SR, et al. "In vitro e in silico evaluation of the inhibition of Staphylococcus aureus efflux pumps by caffeic and gallic acid." Comparative immunology, microbiology and infectious diseases 57 (2018): 22-28.
Balbino, VQ. Tannic acid affects the phenotype of Staphylococcus aureus resistant to tetracycline and erythromycin by inhibition of efflux pumps. Bioorganic chemistry, 74, (2017) 197-200.
The quality of English writing throughout the manuscript needs minor adjusting. Native or professional English writer assistance may be required.
Author Response
"Please see the attachment."

Reviewer 2 Report
The manuscript from Sykuła et al. describes the development and characterization of new derivative of hesperetin HABH (2-amino-N'-(2,3-dihydro-5,7-672 dihydroxy-2-(3-hydroxy-4-methoxyphenyl)chromen-4-ylidene)benzohydrazide) and its copper complex CuHABH. The antimicrobial and antibiofilm activities were also evaluated.
The study was well conducted and is well presented. After minor adjustments it will be suitable for publication. The suggestions are provided in the attached pdf file.

Author Response
"Please see the attachment."

Reviewer 3 Report
Dear Author, I reviewed the manuscript entitled: From physicochemical characteristics of novel hesperetin hydrazone to in vitro antimicrobial aspects. This manuscript presents relevant information about herperetin antimicrobial potential. However, some sections of this manuscript can be improved. For this reason, I considered that this manuscript needs major changes for being considered for its publication in this journal.
Additional Comments:
Check typo in the title and sections of this manuscript.
Highlight the advantages of using hesperetin as an antimicrobial compound.
Check paragraphs extension in this manuscript.
Include a bibliographical reference in circular dichroism protocol.
Include an experimental design that contents statistical factors and variables of response in the statistical analyses applied to the findings of this research.
Try to compare the obtained findings with similar assays where similar flavonoids were used as antibiofilm treatment.
Try to discuss a possible hesperetin mode of action against quorum sensing as an anti-virulence response.
Include microscopically scale in figure 6 images.
Include future trends to keep working with the obtained data.
Try to conclude with a general statement of the most relevant part of this study.
Author Response
"Please see the attachment."

Round 2
Reviewer 3 Report
Dear Author, I reviewed the revised version of the manuscript (molecules-1551714) entitled: From physicochemical characteristics of novel hesperetin hydrazone to in vitro antimicrobial aspects. This version of the manuscript followed all the suggested modifications and recommendations by the reviewers. Besides, the findings obtained in this research are well described and compared with bibliographical references and justify the importance of this obtained data. For this reason, I considered that this manuscript can be accepted for publication in this journal.